# The Titles of the High Priest of Jerusalem in the Pre-Hasmonean Period

## Christophe Nihan

Protestant Faculty of Theology, University of Münster, 48143 Münster, Germany; christophe.nihan@uni-muenster.de

**Abstract:** This article studies the main titles documented for the high priest of Jerusalem in the Hebrew Bible as well as in a few other sources from the Persian and Hellenistic periods. In dialogue with recent scholarship on the topic, particularly an important article by Noam Mizrahi it argues that the title הכהן הגדול (*ha-kohēn ha-gādôl*), "high" or "great priest" probably originates in the late monarchic period (seventh century BCE), but only became the standard designation for the high priest during the fifth century BCE. An alternative title, כהן הראש (*kohēn ha-ro'š*), "head" or "chief" priest, was introduced in Chronicles and other writings in order to designate the high priests of the preexilic period specifically. Finally, a third title, המשיח (*ha-kohēn ha-māšîah*), "the anointed priest", was used for some time in priestly circles as part of a bid to transfer a key royal attribute (anointment) to the high priest of Jerusalem, but was eventually replaced with the more standard designation הכהן הגדול.

**Keywords:** high priest; Judean priesthood; Persian period; Hellenistic period

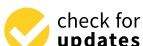

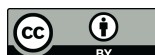

The high priest was the chief of the local cult in Jerusalem and Samaria and, as such, a figure of major social, religious, and political importance. In the case of the high priest of Jerusalem,[1] various titles are documented for the pre-Hasmonean period in the Hebrew Bible and, to some extent, in epigraphic and literary sources. While it is commonly agreed that these titles provide some significant information about the history of the high priest and the way in which this figure was construed, several questions regarding their origin and significance remain in need of clarification. A major step in the recent discussion is provided by the article by Noam Mizrahi (Mizrahi 2011), which basically seeks to align the distinction between the two main titles for the high priest in the pre-Hasmonean period, with the distinction between "Early" and "Late" Biblical Hebrew (EBH and LBH, respectively). However, Mizrahi's analysis is not without problems, as we will see below. Further clarity in these issues can only be achieved through a careful assessment of the textual and literary complexities involved in the passages where the titles of the high priest are found.

## 1. The Evidence Surveyed

The Hebrew Bible documents two main titles for the high priest of Jerusalem: הכהן הגדול (*ha-kohēn ha-gādôl*), "high" or "great priest", and כהן הראש (*kohēn ha-ro'š*), literally "head priest" or "chief priest". The first title, הכהן הגדול, is the most frequent one and occurs twenty times in total in the Masoretic text (MT) of the Hebrew Bible: twice in the Pentateuch, once in Joshua, four times in Kings, five times in Haggai, three times in Zechariah, three times in Nehemiah, and once in Chronicles (see Num 35:25, 28; Josh 20:6; 2 Kgs 12:11; 22:4, 8; 23:4; Hag 1:1, 12, 14; 2:2, 4; Zech 3:1, 8; 6:11; Neh 3:1, 20; 13:28; 2 Chr 34:9); in addition, the phrase הכהן הגדול is already found once in Lev 21:10, where it is not used as a title, but embedded in a syntactic construction which describes the high priest's function (more on this below). The title is not attested in epigraphic sources from pre-Hasmonean times. However, in one passage of the Elephantine papyri, the high priest of Jerusalem is designated with an expression, רבא כהנא (TAD A 4.7:18; for the edition see Porten and Yardeni 1993, pp. 69–70), which is the Aramaic equivalent of the Hebrew compound הכהן הגדול, and is arguably modeled on it (for a detailed discussion, see Mizrahi 2011, pp. 693–96). The

second expression, כהן הראש, occurs less frequently in the Hebrew Bible; it is found mostly in Chronicles (2 Chr 19:11; 24:11; 26:20), as well as once in Kings and Jeremiah, in a parallel passage (2 Kgs 25:18 // Jer 52:24). Additionally, the variant form הכהן הראש (with the article before כהן "priest") occurs once in Ezra 7:5 and in 2 Chr 31:10, whereas the form הכהן ראש is likewise attested once in 1 Chr 27:5.

Aside from these two designations, a further compound, הכהן המשיח (*ha-kohēn ha-māšîah*), literally "the anointed priest", is attested a few times in the Pentateuch, mainly in Lev 4, the legislation on the חטאת, or "purification", offering (see Lev 4:3, 5, 16; further Lev 6:15; Num 3:3, where the expression is used in the plural, הכהנים המשחים, to designate the sons of Aaron in general rather than the high priest specifically). Finally, in many other instances, the high priest is merely designated with the expression הכהן (*ha-kohēn*), "the priest". This usage is especially prominent in the Pentateuch and Joshua where the title הכהן is usually specified by its association with the name "Aaron" (or "Eleazar" in Numbers and Joshua, following the death of Aaron recounted in Num 20:22–29), so that no further title seems to be required in order to identify the high priest (see the construct אהרן הכהן, "Aaron the priest", in Ex 31:10; 35:19; 38:21; 39:41; Lev 1:7; 7:34; 13:2; 21:21; Num 3:6, 32; 4:16, 28, 33; 7:8, 12; 18:28; 25:7, 11; 26:1, 64; 33:38; Josh 21:4, 13; compare also Ezra 7:5). Outside of the Hexateuch, this usage is mainly found in the passages of Kings (see 2 Kgs 11:9, 10, 15, 18; 12:3, 8, 10, 11 [Jehoiada]; 16:10, 11, 15, 16 [Uriah]; 22:10, 12, 14; 23:24) and Chronicles (1 Chr 16:39; 24:6; 2 Chr 22:11; 23:8, 9, 14; 24:2, 20, 25; 34:14, 18), but seldom outside of these two books[2]. Furthermore, the same usage appears to be reflected in a coin from the early Hellenistic period, inscribed with the legend in paleo-Hebrew YWHNN HKHN, "Yohanan the priest" (for the initial publication, see Barag 1986–1987). A recent analysis of the series to which this coin belongs indicates that it shortly follows the introduction in Judea of the Attic weight by the Macedonians and should be dated accordingly to the end of the fourth century BCE (Gitler and Lorber 2008, pp. 69–70). While there has occasionally been some discussion concerning the meaning of the title HKHN in this coin (see Mildenberg 1988, pp. 724–25; further Grabbe 1994, p. 71, who suggest that הכהן may designate a functionary from a priestly family), it is generally agreed now that this title must refer to the high priest of Jerusalem (e.g., Fried 2004, pp. 227–31; Oswald 2015, p. 311: "Der hier genannte Priester kann niemand anders als der Hohepriester sein [ . . . ]"; Monson 2016, p. 18), who presumably became the minting authority in Jerusalem following the disappearance of the office of governor under Ptolemy I.[3]

Two preliminary remarks can be made regarding the evidence presented here. Firstly, while the distribution of the titles for the high priest is a complex phenomenon, some general trends can nonetheless be observed. In particular, the (rare) title הכהן המשיח, "the anointed priest", is exclusively found in "Priestly" texts of the Pentateuch (on the origin of which, see below). The title כהן הראש and its variants (הכהן ראש and הכהן הראש), for their part, are never found in the Pentateuch, and mainly occur in Chronicles (five out of eight occurrences). As far as this is possible, a comprehensive interpretation of the titles of the high priest in the pre-Hasmonean period should seek to account for such patterns. Secondly, the question of the extent to which these titles were effectively used in order to designate the high priest of Jerusalem during the pre-Hasmonean period is a difficult one, and arguably requires a differentiated answer. The only title for the high priest which is securely documented outside of the Hebrew Bible is הכהן, since it is mentioned in the legend of the paleo-Hebrew fractal coin from the end of the fourth BCE, as discussed above. Additionally, the expression כהנא רבא in the Elephantine letter TAD A 4.7:18 to designate the high priest of Jerusalem seems to be an Aramaic calque of the Hebrew title הכהן הגדול. If so, and since the letter itself dates from 407 BCE, we may legitimately infer that, by the end of the fifth century BCE, the latter title was not only used in Yehud, but also relatively well-known by some Jewish communities outside of the province. Whether, and to what extent, the alternative title כהן הראש was effectively used in the pre-Hasmonean period is less obvious. As we will see, there are good reasons to consider that the title is, in fact, a scribal construct and was never applied to a high priest during his office. The existence of

variant formulations, such as הכהן הראש and הכהן ראש, appears to indicate at any rate that it was not (yet) a frozen designation for the high priest of Jerusalem. A similar conclusion applies in the case of הכהן המשיח; its distribution suggests that it may have been used predominantly, or even exclusively, in priestly circles, and perhaps only for a limited period of time (see further below).

## 2. The Evidence Interpreted

Even if we take into account the uncertainties noted above concerning the use of these titles, there is general agreement among scholars that the variety of titles evinced by the Hebrew Bible and, to some extent, other sources, points to some sort of diachronic development. This inference is consistent with the basic observation that these titles occur in different collections, and are written by different authors in different periods. Furthermore, the fact that in at least one instance (2 Chr 24:11), the rewriting of a passage of Kings in Chronicles goes together with the replacement of one title with another (see below) appears to support the view that the use of these titles was dynamic rather than static. The question, then, is what model(s) of diachronic development for the titles of the high priest can legitimately be generated on the basis of the evidence surveyed above.

Until recently, the dominant scholarly view was that הכהן הגדול should be viewed as a late innovation, postdating the second capture of Jerusalem by the Babylonian army in 587 BCE and the (at least partial) destruction of the city's temple. On the other hand, the more seldom title כהן הראש was often assumed to preserve an earlier usage, going back to the monarchic period (see, e.g., De Vaux 1960, pp. 241–42; Cody 1969, pp. 103, 176; Rooke 1998, p. 194 with n. 20, 201–2; Rooke 2000, p. 73 n. 156, with additional references). D. Rooke, in particular, even suggested that the term ראש in the title כהן הראש initially referred to the king, so that this compound would have designated the chief-priest of the Jerusalem temple in his quality, or competence, as a surrogate for the king ("the priest of the head = king"; see Rooke 1998, pp. 194–97; but contrast, e.g., Bartlett 1969, pp. 5–6). Despite its wide reception, this longstanding scholarly view regarding the relative chronology of the two main titles for the high priest is not without issues, especially when one considers that most occurrences (five out of eight) of the title כהן הראש are found in Chronicles, a rewriting of Kings from the Late Persian or Early Hellenistic period (fourth/third century BCE).

The latter point is precisely one of the arguments developed by Mizrahi in an article where he challenges, and even reverses, the dominant view (Mizrahi 2011). For Mizrahi, it is הכהן הגדול which corresponds to the earlier, monarchic title of the high priest, whereas כהן הראש reflects a later, postexilic development. As a matter of fact, regarding the notion that הכהן הגדול rather than כהן הראש would reflect the title of the high priest in the monarchic period, a similar claim was already made by scholars, such as Menahem Haran or Jacob Milgrom, who regarded the "Priestly" texts of the Pentateuch or Hexateuch in which הגדול הכהן occurs (Lev 21:10; Num 35:25, 28; Josh 20:6) to be preexilic rather than postexilic in origin (see Haran 1978, pp. 93–94; Milgrom 2000, pp. 1812–13). The main novelty in Mizrahi's argument, at least as far as the dating of these titles is concerned, resides in the attempt to match the distinction between הכהן הגדול and כהן הראש with the distinction between "Early" and "Late" Biblical Hebrew. According to Mizrahi, הכהן הגדול fits well with the linguistic patterns of Early Biblical Hebrew (EBH), whereas the alternative title כהן הראש shows several signs of belonging to a later stage in the development of Hebrew language, or Late Biblical Hebrew (LBH). Essentially, Mizrahi's argument for the postexilic origin of the title כהן הראש is based on three observations (Mizrahi 2011, pp. 688–93). Firstly, and consistent with point noted above, he observes that most of the attestations of the title כהן הראש (or its variants) are found in books that are clearly postexilic (Ezra and Chronicles), whereas the two remaining attestations, in 2 Kgs 25 and Jer 52, cannot predate the Neo-Babylonian period since they relate to the capture and destruction of Jerusalem. Secondly, Mizrahi rightly notes that in one instance, 2 Chr 24:11, the text of Chronicles reproduces Kings (2 Kgs 12:11) but replaces the title הכהן הגדול in Kings with כהן הראש (Mizrahi 2011, p. 690). Finally, Mizrahi remarks that the existence of the variant form הכהן הראש, in which

the *nomen regens* also takes the definite article (Ezra 7:5; 2 Chr 31:10), reflects a phenomenon which is typical of LBH (Ibid.: 691–92).

From these observations, Mizrahi concludes that the compound כהן הראש is typically LBH, and that it was meant to replace the earlier, preexilic title הכהן הגדול for the high priest. According to him, the meaning of the term ראש in this context would be primarily genealogical, rather than administrative; namely, its function would have been to emphasize that the high priesthood belongs to the eldest son and "to mark the genealogical hierarchy of the priests within their family tree" (Mizrahi 2011, p. 701). Additionally, Mizrahi argues that this use of ראש in the compound כהן הראש is consistent with a larger lexical shift in postexilic times, when the earlier pair comprising בכור and משנה denoting the first (eldest) and second son, respectively, was replaced by the pair comprising ראש and משנה (Mizrahi 2011, pp. 699–701). He also suggests that the need for this new designation emerged in a historical context when the meaning of the former title הכהן הגדול was disputed, and could apparently be interpreted as referring to competence, or valor, rather than age. In this historical context, the meaning and function of the new designation כהן הראש would have been to reassert the (presumably traditional) notion that the legitimate high priest is the firstborn son (Ibid., pp. 703–5).

In this author's view, Mizrahi's arguments regarding the origin of the title כהן הראש in postexilic times are cogent. With regard to the parallel passage in 2 Kgs 25:18 and Jer 52:24, which—as already noted—is the only occurrence of כהן הראש outside of the books of Ezra and Chronicles, it may be added that recent discussion has shown that two texts have a complex redactional history. Their development is still partly documented by the comparison between the Old Greek and Masoretic versions of Jer 52 (=Jer 39 LXX) with 2 Kgs 25, and presumably extends down into the Hellenistic period (on this issue, see now Ammann 2021 with further bibliography). The presence of the title כהן הראש, otherwise attested only in Ezra and Chronicles, is therefore consistent with the notion that the account in 2 Kgs 25 and Jer 52 has undergone substantial revision during the postexilic period (as noted, e.g., by Wöhrle 2008, pp. 218–19)[4]. As such, there is really no reason to consider that כהן הראש could preserve a preexilic usage, and this old theory should now be definitely abandoned.

Other aspects of Mizrahi's argument are, however, more questionable. Two issues, in particular, deserve further discussion in my opinion. The first issue concerns the relative chronology of the titles הכהן הגדול and כהן הראש as well as, more generally, Mizrahi's attempt to match the distinction between these titles with the distinction between EBH and LBH. The fact that Mizrahi can successfully demonstrate that כהן הראש belongs to LBH does *not* prove yet that הכהן הגדול must be substantially earlier and reflect the usage in EBH. As a matter of fact, a large number of texts where the title הכהן הגדול occurs appear to be similarly late compositions and/or to have undergone editorial revisions. From a historical perspective, this observation raises the question of whether, and to which extent, these texts can then be successfully used in order to retrieve high priestly titles from the monarchic period. A second, distinct issue, concerns the explanation offered by Mizrahi for the introduction of the title כהן הראש at some point during the postexilic period. Despite the evidence adduced by Mizrahi, the argument according to which the introduction of כהן הראש would seek to reaffirm the genealogical dimension of the high priestly office in a social-historical context where the traditional understanding of this office was disputed relies on a problematic basis, not the least because (a) there is no evidence that כהן הראש points more clearly to a genealogical understanding of the high priestly office; and (b) there is likewise no compelling evidence that the genealogical understanding of the title הכהן הגדול was in question during the pre-Hasmonean period. The following discussion will address each of these issues in turn. In addition, it will also discuss the origin and *raison d'être* of a third title for the high priest, הכהן המשיח, an issue which Mizrahi left unaddressed in his 2011 article.

### 3. The Title הכהן הגדול in the Hebrew Bible: A Reexamination

Out of the twenty occurrences of the title הכהן הגדול, twelve are found in books which are clearly postexilic: Haggai, Zechariah, Nehemiah, and Chronicles. This observation already points to a basic issue involved in the attempt to match the distinction between הכהן הגדול and כהן הראש with the distinction between EBH and LBH. Not only did הכהן הגדול remain in use in postexilic times, but there are in effect more attestations of this title in postexilic texts than the alternative title כהן הראש. While this conclusion does not preclude the possibility that הכהן הגדול originates in the monarchic period, it shows that the chronological relationship between the two titles is complex and that both titles could in effect coexist during the postexilic period. As a matter of fact, Mizrahi rightly observes that the alternative title כהן הראש was short-lived during the postexilic period (2011, pp. 692–93); it is only seldom found in Qumran Hebrew and has entirely disappeared in Mishnaic Hebrew, where it is consistently replaced with the title הכהן הגדול. As for the view that הגדול הכהן was the title of the high priest in Jerusalem during the preexilic period, the evidence for or against it rests entirely on the eight remaining occurrences found in Leviticus (21:10), Numbers (35:25, 28), Joshua (20:6), and 2 Kings (12:11; 22:4, 8; 23:4). Mizrahi is aware of this issue (see his brief comment in 2011, p. 694 n. 28), but does not discuss it further. Yet in several instances, a preexilic origin for these texts appears to be questionable, to say the least.

In the Pentateuch, the first occurrence of הכהן הגדול is found in Lev 21:10, a text which belongs to the Holiness legislation (H). The Holiness legislation is now widely recognized to be post-Priestly and to date from the Neo-Babylonian period at the earliest (Grünwaldt 1999; Stackert 2007; Carr 2011, pp. 289–303; Schmid 2012, pp. 176–77; other scholars, such as I. Knohl and J. Milgrom, date the composition of H to the preexilic period, but nonetheless acknowledge that it continued to be supplemented down to the exilic or, for Knohl, even the post-exilic period: see Knohl 1995; Milgrom 2000, pp. 1361–364), although a dating in the Persian period, probably in the fifth century BCE, seems more likely (Otto 1994; Nihan 2007, pp. 545–59; Schmid 2012, pp. 176–77). That H which postdates the exile (at least in its present form) is already shown by its conclusion in Lev 26, which refers not only to the exile itself (26:36–39), but even to the return from exile (26:40–45; compare Deut 30:1–10). This conclusion is further supported by H's phraseology, which borrows not only from P and Deuteronomy, but also from various prophetic texts from the exilic and postexilic periods (e.g., Müller 2010; Nihan 2022). The title הכהן הגדול is used again twice in the legislation of Num 35:9–34 on cities of refuge in case of inadvertent homicide (v. 25 and 28). Num 35 has long been shown to be a late composition within the book of Numbers, which already presupposes the Holiness legislation (e.g., Achenbach 2003, pp. 598–600; Frevel 2013, pp. 159–60). This conclusion likewise applies to the designation of the high priest with the title הכהן הגדול, which is modeled on Lev 21:10 (Frevel 2013, pp. 159–60). Finally, the legislation of Num 35 on cities of refuge has a counterpart in Josh 20, where we find the fourth and last occurrence of the title הכהן הגדול in the Hexateuch (v. 6). As various authors have shown, Josh 20 presupposes the legislation of Num 35 and was apparently devised as a supplement of sorts to this legislation (van der Kooij 1997; Stackert 2007, pp. 96–111). Moreover, v. 4–6 are missing from the Codex Vaticanus (G[B]), which arguably represents the Old Greek text of Josh 20 (see Müller et al. 2014, pp. 45–58)[5]. In this case, therefore, the late origin of this passage is corroborated by the textual evidence. In short, the four passages mentioning the title הכהן הגדול in the Hexateuch are closely interconnected and reflect a diachronic development, the origin of which is to be found in Lev 21:10 (Lev 21:10 → Num 35:25, 28 → Josh 20:6). Since Lev 21:10 itself belongs to the postexilic Holiness legislation, none of these four passages can provide reliable evidence for the origin of this title in the monarchic period.[6]

What remains, therefore, are the four occurrences of the title הכהן הגדול preserved in Kings. Of these four occurrences, one is found in the account of Jehu's repairs of the temple in 2 Kgs 12:5–17 (12:11), whereas the remaining three are located in the account of Josiah's so-called "reform" in 2 Kgs 22–23 (22:4, 8; 23:4). The account of Jehu's repairs of the temple

is commonly viewed as a late composition, derived from the parallel account in 2 Kgs 22:4–10 (see, especially, Spieckermann 1982, pp. 179–83; Levin 2003, pp. 169–97; compare also, e.g., Würthwein 1984, pp. 354–57, who regards 2 Kgs 12:5–17 as a post-dtr addition). It is unlikely, therefore, to belong to the early stages in the edition of 1 and 2 Kings. Furthermore, the use of the compound הכהן הגדול in 2 Kgs 12:11 is problematic, because everywhere else in 2 Kgs 11–12 Jehoiada is simply designated with the title הכהן "the priest" (compare 2 Kgs 11:9, 10, 15, 18; 12:3, 8, 10). Since 2 Kgs 12:11 is the *last* passage where Jehoiada is effectively mentioned, it is possible that the sudden mention of his title as הכהן הגדול, and not just הכהן, goes back to a later editor, as S.L. McKenzie has recently argued. This editor wanted to make clear that Jehoiada, as the chief-priest of the Jerusalem temple in Jehu's reign, had the same status and dignity as the high priests of later, postexilic times (McKenzie 2019, p. 457: "an anachronistic gloss"; compare already De Vaux 1960, p. 242). If this is so, the occurrence of the compound הכהן הגדול in 2 Kgs 12:11 may in fact reflect a late scribal gloss from the Second Temple period.

The case of the three occurrences of הכהן הגדול in the account of 2 Kgs 22–23 is more difficult. The account has a complex history, which cannot (and need not) be discussed in detail in the limits of this article. Of the three occurrences of the title הכהן הגדול, the one in 22:8 is unlikely to belong to the original account. V. 8 has long been identified as a later addition within the account of the temple's repair in 22:4–10, which interrupts the narrative sequence between v. 7 and 9 and serves to introduce the motif of the finding of the ספר התורה, the "Book of the Law" (see, e.g., Levin 2003, pp. 213–14; Pakkala 2010, p. 225 with n. 75). The same conclusion applies to the third occurrence in 23:4, a verse which is located at the joint between the account of the repairs of the temple instructed by Josiah in 22:3–7, 9 and the account of the cultic reform in 23:4–20. Even if we assume that the account of Josiah's reform in 22:4–20—or, rather, a portion of it—was part of the earliest account in 2 Kgs 22–23 (which is not unanimously accepted: compare, e.g., Levin 2003, pp. 198–216), it is unlikely that it included v. 4. The language of v. 4a, which mentions the worship of "Baal, Ashera, and all the Host of Heavens", is exclusively found in a few late passages of Kings and Chronicles, namely, 2 Kgs 17:16; 21:3, and 2 Chr 33:3. From a religious-historical perspective, the association of Ashera with Baal and other astral deities (the "Host of Heavens") reflects the later, postexilic polemics against these deities, not the preexilic practice (see, e.g., Spieckermann 1982, pp. 79–83; Hentschel 1985, p. 110: "Die Göttertrias 'Baal, Aschera und das ganze Heer des Himmels' verrät den Eingriff eines dtr Redaktors"). The identification of v. 4a as a later addition accounts for the fact that the high priest, who is mentioned in this half-verse, effectively plays no role whatsoever in the subsequent narrative of the cult reform.[7] Based on these observations, it seems almost certain that the occurrence of הכהן הגדול in 2 Kgs 23:4a was not an integral part of the narrative, but reflects a later revision from postexilic times.

What remains, then, is the occurrence of the title הכהן הגדול in 2 Kgs 22:4. In this case, it is indeed possible that this verse provides evidence for the use of this title in the (late) monarchic period. V. 4 forms an integral part of the "repair" account in 22:4–10*, which is itself unanimously recognized as comprising the earliest layer in the narrative of 2 Kgs 22–23. It has sometimes been argued that the mention הגדול after הכהן would betray a later revision (e.g., De Vaux 1960, pp. 241–42; Spieckermann 1982, p. 47 n. 33; Würthwein 1984, p. 446 n. 3). While this is possible in principle, this solution is not supported by the text-critical evidence and remains strictly hypothetical. More likely, therefore, 2 Kgs 22:4 may in fact represent the earliest attestation of the title הכהן הגדול in the Book of Kings (as correctly identified by Levin 2003, p. 189 with n. 108) and, by extension, in the Hebrew Bible as a whole. The origin of the account in 2 Kgs 22:4–10* remains unclear. As various authors have argued, it is likely that this account goes back to a source used by the Deuteronomistic (Dtr) redactors who composed the account of Josiah's reform (see, especially, Spieckermann 1982, p. 183; Levin 2003, pp. 188–89; most recently Pakkala 2010, pp. 225–26, who concludes: "Consequently, it is probable that 2 Kings 22:3–7, 9 was the spark and foundation of Josiah's reform and already an integral part of the history writer's

text, most likely taken from one of his sources"). In particular, this reconstruction can satisfactorily account for the divergencies that can be observed between the accounts of the temple's repair in 22:4–10 and of Josiah's cultic reform in 23:4–20. If this analysis is correct, 2 Kgs 22:4 would effectively provide evidence, albeit limited, for the use of the title הכהן הגדול in the late monarchic period (seventh century BCE) already. Quite possibly, the introduction of this title for the chief priest of the cult in Jerusalem during the late monarchic period may reflect an attempt to imitate Akkadian titles for high-ranking priests that were still used in the Neo-Assyrian period, such as *šangû rabû*—of which the Hebrew הכהן הגדול is a fairly exact equivalent (for the equivalence between Akk. *šangû* and Hebrew *kohēn*, see Cody 1969, pp. 100–2). However, the evidence provided by 2 Kgs 22:4 is too limited to warrant any solid conclusion in this regard.

## 4. The Title כהן הראש in the Postexilic Period: Toward an Alternative View

　　While the title הכהן הגדול may originate in the late monarchic period, the alternative title כהן הראש is exclusively documented in postexilic texts and appears, therefore, to be a creation from the postexilic period. If so, this conclusion raises the question of the motivations underlying the emergence of this alternative title. As noted above, Mizrahi—who, to my knowledge, is the first to address this question in-depth—develops a sophisticated argument according to which the introduction of כהן הראש would seek to reaffirm the primarily *genealogical* dimension of the high priestly office. Mizrahi's argument in this regard is primarily based on the biblical evidence, but it also includes some non-biblical sources regarding the high priest during the Persian and Hellenistic periods. On closer examination, however, none of the evidence discussed by Mizrahi provides sound support for his interpretation of the emergence of כהן הראש as an alternative title for the high priest. Specifically, two sets of observations may be advanced here.

　　(1) Regarding the use of the compound כהן הראש in the Hebrew Bible, there is hardly any evidence supporting the view that the meaning of this compound would be primarily genealogical. Mizrahi is certainly correct that the pair משנה/בכור denoting the eldest and second son, respectively, came to be replaced in LBH by the pair משנה/ראש. However, this contrast never seems to be operative in the case of the high priest. In the parallel account of 2 Kgs 25:18 and Jer 52:24, where כהן הראש is used in contrast with כהן משנה (2 Kgs 25:18), or כהן המשנה (Jer 52:24), the distinction seems clearly *administrative* rather than genealogical. It points to the hierarchy that prevails between the chief-priest of Jerusalem, Seraiah, and the priest-in-second, Zephaniah (Rooke 1998, p. 197 rightly speaks in this regard of an "evidently hierarchical arrangement"). There is no indication that the two priests are brothers: the book of Jeremiah consistently presents Zephaniah as "the son of Maaseiah" (Jer 21:1; 29:25 MT = 36:25 LXX), whereas according to 1 Chr 6:14, Seraiah was the son of Azariah.[8] Furthermore, the expression כהן משנה occurs in one more passage, 2 Kgs 23:4, where it is used in the plural, כהני המשנה, denoting several second-ranking priests, and in combination with הכהן הגדול (and not with כהן הראש!). This finding is consistent with the view that all these titles are administrative rather than genealogical. Additionally, the fact that משנה כהן is used jointly with הכהן הגדול in this passage undermines the idea that כהן הראש was specifically introduced to form a contrast with כהן משנה, emphasizing the genealogical preeminence of the high priest over other male members of his family.

　　The other passages where the expression כהן הראש, or one of its variants, occurs do not provide further evidence for Mizrahi's interpretation. On the contrary, the administrative meaning of the title כהן הראש is manifest in some passages, such as 2 Chr 19:11a (unless otherwise specified, all translations are from the author):

<div dir="rtl">

והנה אמריהו כהן הראש עליכם לכל דבר יהוה

ובדיהו בן ישמעאל הנגיד לבית יהודה לכל דבר המלך

ושטרים הלוים לפניכם

</div>

　　See, Amariah the high priest is over you for all of Yhwh's matters;

and Zebadiah son of Ishmael, the governor of the house of Judah, for all of the king's matters;

and the Levites will serve you as officers.

There is a clear parallel established between Amariah, who is in charge of "all of Yhwh's matters", and Zebadiah, who is in charge of "the king's matters". Furthermore, both characters are designated with the title corresponding to their function: "Amariah, the high priest" (כהן הראש) and "Zebadiah, the governor of the house of Judah" (הנגיד לבית יהודה); however, while Zebadiah's ascendency is provided ("son of Ishmael"), there is no comparable mention for Amariah. The parallel with the title הנגיד לבית יהודה, combined with the absence of genealogy, clearly suggests in this case that כהן הראש is used to denote an administrative function, or position, as scholars generally recognize.

(2) Mizrahi's theory regarding the emergence of כהן הראש as an alternative title for the high priest during the postexilic period is predicated upon a fairly complex historical scenario, according to which the attribution of the high priesthood to the eldest son became disputed in postexilic times. However, the evidence that Mizrahi can muster in support of this scenario is problematic. Mizrahi begins this part of his discussion with a reference to Lev 21:10a, the first passage in the Pentateuch where the title הכהן הגדול is introduced. In this passage, the title הגדול הכהן is immediately qualified with the phrase מאחיו, which functions as an explicative comment of sorts: "the priest who is greater than his brothers"; it is not entirely clear, however, to what sort of superiority the text refers here (more on this below). Mizrahi reads this passage in the light of a later interpretation found in a Tannaitic tradition (preserved in *t. Kipp.* 1:6 and *Sifra 'Emor* 2:1), according to which the statement in Lev 21:10 means that the high priest must be "greater than his brothers in beauty, strength, wealth, wisdom and appearance" (Mizrahi 2011, p. 702). He also remarks that a similar view of the high priest is preserved in a passage of Diodorus of Sicily, from Book 40 of his *History*, the "Excursus on the Jews", which was classically attributed to Hecateus of Abdera (c. 300 BCE) and where the high priest is described as "superior to his colleagues in wisdom and virtue" (φρονήσει καὶ ἀρετῇ προέχειν)[9].

> διὸ καὶ βασιλέα μὲν μηδέποτε τῶν Ἰουδαίων, τὴν δὲ τοῦ πλήθους προστασίαν δίδοσθαι διὰ παντὸς τῷ δοκοῦντι τῶν ἱερέων φρονήσει καὶ ἀρετῇ προέχειν. τοῦτον δὲ προσαγορεύουσιν ἀρχιερέα, καὶ νομίζουσιν αὐτοῖς ἄγγελον γίνεσθαι τῶν τοῦ θεοῦ προσταγμάτων.

> For this reason the Jews never have a king, and authority over the people is regularly vested in whichever priest is regarded as superior to his colleagues in wisdom and virtue. They call this man the high priest, and believe that he acts as a messenger to them of God's commandments.

<div align="center">Photius, <em>Bibl</em>. Cod. 244 (381a) = Diodorus, 40, 3, 5 (my translation)</div>

Finally, Mizrahi (2011, pp. 703–5) references the evidence from Josephus for the internal struggles between brothers over the priesthood during the Persian period, namely, Yohanan and Joshua (*Ant.* 11.297–301) as well as Jadduah and Manasseh (*Ant.* 11.302–346).

There are, however, several issues with those sources. The attribution of the "Excursus on the Jews" to Hecateus of Abdera was based on a fairly traditional view of Diodorus as a mere compiler of sources, which has been substantially challenged. Accordingly, more recent studies emphasize Diodorus' role as an author rather than as a compiler (see, especially, Muntz 2017, pp. 1–26, with additional references). Furthermore, as various scholars have shown, the ascription of the Excursus—which is known to us only through the paraphrase of Photius in the ninth century CE—in its entirety to Hecateus of Abdera is unlikely for several reasons, not the least because the Excursus provides several details which reflect a later historical context than the time of Ptolemy I (see, especially, Schwartz 2003; Gmirkin 2006, pp. 38–71; Zamagni 2010; and most recently Kratz 2021, pp. 271–74). Although Diodorus is likely to have used various sources himself—as would be expected from an ancient historian—the nature and extent of these sources remains quite difficult

to identify. Therefore, as C. Zamagni (2010) has cogently argued, the Excursus should be regarded as a source of Diodorus, not of Hecateus. Last but not least, the description of the Judean high priest as "superior to his colleagues in wisdom and virtue" follows a typically Greek pattern in the representation of leadership; as rightly noted by Bar-Kochva (2010, pp. 124–25), φρόνδησις and ἀρετή "are characteristics the average Greek would expect a leader to have". As such, Diodorus' description of the high priest emphasizing his moral value rather than his genealogy represents an *interpretatio graeca*; it does not provide a reliable source to understand the *Judean* representation of the high priest during the Hellenistic period. Admittedly, the Tannaitic tradition cited by Mizrahi suggests that at some point during the Early Roman period (or slightly earlier), this *interpretatio graeca* found a limited reception in the Jewish tradition of the high priest. However, to project this tradition back into the period of the composition of the biblical texts mentioning the title הכהן הגדול seems adventurous.

Likewise, Josephus' reports about brotherly struggles for the high priesthood cannot prove Mizrahi's point. The question of the extent to which these two accounts go back to earlier sources used by Josephus and may preserve historical information is complex, and cannot be discussed here at length. Regarding *Ant*. 11.302–346, Josephus' account of the founding of the Samaritan sanctuary on Mount Gerizim is contradicted by archaeological evidence, which indicates that this sanctuary was established not in the time of Alexander's conquest of the Levant, but significantly earlier, toward the middle of the fifth century BCE (see Magen 2000, 2007). While this does not automatically preclude the possibility that Josephus' account preserves some historical details, it certainly suggests that the story of the conflict between the two brothers, Jadduah and Manasseh, needs to be apprehended with caution, and may be more legendary than historical. In the case of the first account (*Ant*. 11.297–301), various authors have argued that the story of Joshua's unsuccessful challenge of his elder brother's claim to the priesthood, and the subsequent murder of one brother by the other, has a historical basis (for instance, Albertz 2011, with additional references). At the very least, the notion that the high priest had to be confirmed by the local representative of the Achaemenid administration corresponds to the situation documented elsewhere in the Achaemenid empire, as rightly pointed out by L.S. Fried on the basis of Egyptian and Babylonian evidence (Fried 2004). However, the account does imply that, *were it not for the active support of the local governor*, Joshua would not have had any grounds to challenge his elder brother's claim to the title of high priest. In other words, the account presupposes a situation in which it was expected for the elder son to become high priest after his father. Far from challenging the traditional mechanism of high priestly succession, the account—if it is historical—actually confirms that this mechanism was the norm in Yehud throughout the Persian period. Note that the same point applies to the account in *Ant*. 11.302–346, even though—for the reasons indicated above—the historicity of this account appears highly questionable; even though Manasseh was supported by the governor of Samaria Sanballat, his father-in-law, it was Jadduah who inherited the high priesthood *because* he was the eldest son.

In short, and summing up the discussion so far: there is no significant evidence to support Mizrahi's claim that the principle of the inheritance of the high priestly office by the eldest son was significantly challenged during the pre-Hasmonean period, and this assumption cannot explain the emergence during Persian and Early Hellenistic times of the alternative title כהן הראש. The interpretation of the title for the high priest in moral rather than genealogical terms reflects an *interpretatio graeca* which is documented for the first time in the work of Diodorus (first century BCE), and which cannot be used therefore to identify *Judean* attitudes toward the high priest in the Persian and Early Hellenistic period. Josephus' accounts of brotherly struggles in the high priestly family during the Persian period are historically questionable sources, especially regarding *Ant*. 11.302–346, which at any rate, presuppose a situation where the transmission of the high priesthood to the elder son was still the norm in Judea. The first documented instance where the genealogical principle in the succession of high priests was effectively challenged is under Antiochus

IV (175–164 BCE), when—in accordance with Greek practice—the high priestly office in Jerusalem was offered to the highest bidder and Jason could take the place of Onias III (2 Macc 4:7–10). Before that time, there is no reason to assume that the genealogical principle in the succession of high priests needed to be reaffirmed.

If we ask, then, why the alternative title כהן הראש was developed at some point, a much simpler explanation presents itself when it is observed that this title is exclusively used to denote high priests before and up to the destruction of the first temple of Jerusalem: Aaron (Ezra 7:5), Jehoiada under king David (1 Chr 27:5), Amariah in the reign of Jehoshaphat (2 Chr 19:11), Jehoiada under Joash (2 Chr 24:11), Azariah in the reign of Uzziah (2 Chr 26:20) and again under Hezekiah (2 Chr 31:10), and finally, Seraiah at the time when Jerusalem was captured for the second time and the temple was destroyed (2 Kgs 25:18 // Jer 52:24). After the destruction of the temple, the title כהן הראש is never used again for the high priests of the postexilic period in the Hebrew Bible: for these high priests, it is exclusively the title הכהן הגדול which is used (see Hag 1:1, 12, 14; 2:2, 4; Zech 3:1, 8; 6:11; Neh 3:1, 20; 13:28; 2 Chr 34:9). As a matter of fact, it is precisely *this* observation which led several earlier scholars to assume that כהן הראש should preserve the preexilic designation for the high priest (see, e.g., [Rooke 1998](#), p. 194 n. 20, who argues that הראש כהן goes back to monarchic times because "the title is always used in the context of pre-exilic subject matter"). If, however, we take seriously the evidence indicating that this title is entirely a postexilic construction, then its introduction appears to reflect a concern to coin a designation for the high priests of the preexilic period specifically. In this interpretation, the title כהן הראש is an archaizing designation. It was presumably never used for a high priest in office, but points to a scribal construct highlighting the distinction between preexilic and postexilic high priests. Specifically, the title כהן הראש is used in Chronicles, but also in Kings and Jeremiah, to refer to the chief priest of Jerusalem in the monarchic period, at a time when this priest—differently from his postexilic successor—was still subordinated to the authority of the (Judean) king.[10] As noted above, the quick decline and eventual abandonment of the title כהן הראש during the Hellenistic and Roman periods suggests that this archaizing designation—and the corresponding distinction between the preexilic and postexilic high priest—was not successful. Later Jewish writers preferred the more standard הכהן הגדול title for the high priest, thereby emphasizing the continuity—rather than the discontinuity—between the First and Second Temple periods.

### 5. From the "Anointed Priest" to the "High Priest" in the Priestly Traditions of the Pentateuch

The last point that remains to be explained are the few occurrences of yet another title for the high priest, הכהן המשיח, "the anointed priest". As noted above, this title, in the singular, is exclusively used in Lev 4 and 6 (Lev 4:3, 5, 16; further Lev 6:15 which is based on Lev 4: see on this [Nihan 2007](#), pp. 256–68, esp. 258–260). Since Lev 4 and 6 both belong to the "Priestly" (P) traditions of the Pentateuch, this finding strongly suggests that we have to do with a title that was used for some time by the priestly circles responsible for these traditions (so, e.g., [Milgrom 2000](#), pp. 1812–813). If one accepts the view that (a) the Priestly narrative is a late exilic or early postexilic composition (e.g., [Schmid 2011](#); [Wöhrle 2012](#)), and that (b) Lev 4 and 6 are not an integral part of this narrative, but later additions to it (for a detailed analysis, see [Nihan 2007](#), pp. 160–98), then the use of the title המשיח הכהן for the high priest can be dated fairly securely to the fifth century BCE.

Within the context of P's narrative, this designation clearly refers to the practice of anointing the high priest at the time of his investiture (Ex 29:7; Lev 8:12; 16:32). The latter passage, Lev 16:32, expressly refers to the high priest as הכהן אשר ימשח אתו ואשר ימלא את ידו לכהן תחת אביו, namely, "the priest who is anointed and ordained to act as high priest in place of his father". In all likelihood, this designation for the high priest of Jerusalem corresponds to the ritual that was effectively performed during the investiture of the high priest, even though evidence for such a ritual remains limited outside of P (see, however, Isa 61:1, which may refer to a priestly figure of sorts; further Dan 9:25–26). In several other

traditions of the Hebrew Bible, however, anointment is presented as an important part of the ceremony by which a new king was inthronized (see, especially, 1 Sam 10:1, 15:1, 17; 16:3, 12, 13; 2 Sam 2:4, 7; 5:3, 17; 12:7; 19:10; 1 Kgs 1:34, 39, 45; 5:1; 19:15, 16; 2 Kgs 9:3, 6, 12; 11:12; 23:20; Ps 45:7; 89:20)—a notion which is also preserved in the designation of the king as המשיח, the "anointed one" (1 Sam 2:10, 35; 12:3, 5; 16:6; 24:6, 10; 26:9, 11, 16, 23; 2 Sam 1:14, 16, 21; 19:21; 22:51; 23:1; Isa 45:1; Ps 2:2; 18:51; 20:7; 28:8; 84:9; 89:38, 51; 132:10, 17; Lam 4:20). As such, the use of the title הכהן המשיח for the high priest may reflect, in the Persian period, an attempt to transfer on the high priest one of the key attributes of the king (as argued, e.g., by Gosse 1996), even though such a transfer does not automatically imply the equivalence of the high priest with the king. More limitedly, this development likely suggests that the high priest was susceptible, in a *postmonarchic* context, to take over *some* of the roles and attributes of the king as a communal leader (see the discussion in Nihan 2017, pp. 50–55; Nihan and Rhyder 2018).

At some point, however, this priestly usage was replaced with the more frequent—and arguably better established—title הכהן הגדול, "high priest". Significantly enough, this development appears to correspond to the transition from P to H, the "Holiness" legislation in Lev 17–26, since the title הכהן הגדול is documented for the first time in the Pentateuch in Lev 21:10. The formulation of this passage, which was already mentioned above, is particularly interesting as it clearly represents an attempt to mediate between two traditions for designating the high priest:

Lev 21:10a

והכהן הגדול מאחיו אשר יוצק על ראשו שמן המשחה ומלא את ידו ללבש את הבגדים

The priest who is greater than his brothers, on whose head the anointing oil has been poured and who has been consecrated to wear the vestments . . .

The reference to the high priest as the priest "on whose head the anointing oil has been poured" and "who has been consecrated to wear the (sacred) vestments" corresponds to the conception found in various "Priestly" texts, where the high priest is defined by his anointment (Ex 29:7; Lev 6:13; 8:12; 16:32) as well as by his sacred vestments (Exod 28:2, 3; 29:5, 21, 29; 39:1 MT; 40:13). Exod 29:29, in particular, asserts that, "the holy garments that belong to Aaron are to belong to his sons after him, so that they may be anointed in them and consecrated in them" (translation from NETS). In Lev 21:10, however, the high priest is no longer simply designated as הכהן המשיח, "the anointed priest", but simultaneously with the phrase הכהן הגדול מאחיו, "the priest who is greater than his brothers". As aptly noted by various scholars, this phrase is not a title, but rather a description of the high priest (see, e.g., Milgrom 2000, p. 1812; the alternative view according to which הכהן הגדול מאחיו would represent the high priest's full title, fits neither the syntax of v. 10a nor the Masoretic cantillations, as noted by Milgrom). At the same time, it should be clear that this construction refers to the title הכהן הגדול, which is found elsewhere in the Hebrew Bible (a point which, as far as I can see, is not disputed). This title is explained here through the addition of the phrase מאחיו; namely, the high priest is thus called because he is "greater" than his brothers.

In Lev 21:10, therefore, the traditional Priestly designation of the high priest as the anointed priest is combined for the first time with a reference to the title הכהן הגדול, which simultaneously explains the meaning of this title. The presence of this explanation in Lev 21:10 is indeed fitting, since it is the very first reference to this title in the Pentateuch. For the reasons discussed above (§ 4), there is no need to interpret this re-description of the high priest in a moral or ethical sense, namely, that the high priest would be more virtuous than his "brothers" the priests. More simply, and more in line with the conception of this office not only in the "priestly" portions of Exodus and Leviticus, but also in other books of the Hebrew Bible, the designation of the high priest as הכהן הגדול מאחיו, "the priest who is greater than his brothers", simply denotes the fact that the high priest enjoys a superior rank and status. In other words, the expression הכהן הגדול מאחיו merely expresses the leadership that the high priest enjoys within the priestly class; it is not an ethical concept,

but a political and administrative one. This conclusion is consistent with the occurrence of the construct גדול מן + X in other passages of the Hebrew Bible. In most instances, this construct denotes first and foremost an idea of hierarchical position and power (see, e.g., Exod 18:11; Num 14:12; Deut 1:28; 4:38; 9:1; 11:23; etc.), and this interpretation is likewise fitting in Lev 21:10.

Overall, the transition in the designation of the high priest documented by Lev 21:10 is consistent with the genre and theological aims of the Holiness legislation. As various studies have shown, the Holiness legislation (H) not only postdates a portion of P, but also consistently mediates between priestly and non-priestly traditions (e.g., Stackert 2007; Nihan 2007, pp. 401–545; Carr 2011, pp. 289–303). In the new description provided by H, P's key markers for the high priest—namely, the anointment and the sacred vestments—are maintained, but they are now subsumed to the more standard title הכהן הגדול, which is itself borrowed from non-Priestly traditions outside of the Pentateuch. This development seems to have caused the earlier title הכהן המשיח to become obsolete, since this title is no longer used after H. It only recurs once, in Num 3:3, a late, post-H text in which it is now used in the plural form, הכהנים המשחים, to designate the sons of Aaron in general, and no longer the high priest specifically. This reuse is consistent with another development within the late priestly traditions of the Pentateuch, where anointment is gradually presented as a feature of the Aaronite priests in general, and not just of the high priest (cf. Exod 28:41; 30:30; 40:15; Lev 7:36; 8:10; and see on this Nihan 2007, pp. 589–90).

### 6. Conclusions

The history of the titles of the high priest in the pre-Hasmonean period is a complex and intricated phenomenon, which can only be disentangled through a careful text- and redaction-critical assessment of the various passages within the Hebrew Bible in which these titles are documented. The main findings of the present study can be summarized in the following way.

1. The present analysis has confirmed the view advanced by Mizrahi in his 2011 study, according to which the title הכהן הגדול actually predates the title כהן הראש, the latter being in all likelihood a creation from the postexilic period. Differently from Mizrahi, however, the existence of the two titles cannot be simply aligned with the distinction between "Early" and "Late" Biblical Hebrew: of the twenty occurrences of the title הכהן הגדול only one, in 2 Kgs 22:4, is susceptible of going back to the late monarchic period, while all the remaining occurrences clearly postdate the Neo-Babylonian period.

2. This finding points to a scenario where the title הכהן הגדול may have been introduced at some point during the seventh century BCE as a title for the chief-priest of the cult in Jerusalem, but was consistently used for this figure only during the Persian period, following the rebuilding of the temple and resumption of the regular cult in Jerusalem. The occurrence of the Aramaic equivalent כהנא רבא in one of the Elephantine papyri (TAD A 4.7:18) suggests that הכהן הגדול had become the standard designation for the high priest of Jerusalem by the end of the fifth century BCE.

3. For most of the monarchic period, the chief priest of Jerusalem was merely designated as הכהן, "the priest", a usage which is abundantly documented in 2 Kings and somehow survived in the Pentateuch. Apparently, this designation for the high priest was continued during the Persian and Hellenistic periods and survived alongside the title הכהן הגדול (see also Knoppers 2003), a phenomenon which is confirmed by the fractal silver coin from the end of the fourth century BCE with the legend YWḤNN HKHN, "Yoḥanan the (high) priest".

4. The title כהן הראש, for its part, emerged at some point during the Persian period as an alternative designation for the high priest. Contrary to Mizrahi's view, this development does not appear to reflect the need to emphasize the genealogical dimension of the high priestly office. There is no evidence in biblical and non-biblical sources that the principle of the transfer of the office to the elder son was significantly challenged

before the reign of Antiochus IV in the second century BCE. Rather, the title כהן הראש is used exclusively for the high priests of the preexilic—especially monarchic—period, and therefore, presumably corresponds to the attempt to introduce a historiographical distinction between the preexilic and postexilic high priests. In all likelihood, the title represents a scribal construct and was never used as an actual title for a high priest in office.

5. Finally, a third title, הכהן המשיח, "the anointed priest", was used for some time during the fifth century by the scribes responsible for the composition and transmission of the "Priestly" traditions of the Pentateuch. Since anointment is a key feature of the Israelite and Judean kings in the Hebrew Bible, this title arguably reflects an attempt to transfer a royal attribute to the high priest after the exile and demise of the Judean monarchy. In the context of the merging of the Priestly traditions with other Pentateuchal traditions, however, the designation of the high priest became aligned with the more standard title הכהן הגדול (Lev 21:10, the "Holiness" legislation). Anointment remained a key feature of the high priest in the Priestly and post-Priestly traditions, but it was gradually extended to all Aaronite priests.

6. Overall, this study provides a more nuanced explanation than previous models for the problem of the coexistence of several titles for the high priest within the Hebrew Bible. In particular, the previous analysis confirms that the coexistence of these titles is the result of a diachronic process, which cannot be simply matched with the distinction between the First and Second Temple periods. Rather, the history of each individual title shows a complex chronological overlap: the main title apparently used in the preexilic period, הכהן, continued to be used in the postexilic period, whereas the main postexilic title for the high priest, הכהן הגדול, likely originates in the late Neo-Assyrian period; last but not least, yet another title, כהן הראש, was coined in the postexilic period, but is exclusively used in the Hebrew Bible to designate high priests from the preexilic period. With regard to the history of the institution of the high priesthood, specifically, this result should caution us against overly linear models positing either a strong continuity or a complete break between the preexilic and postexilic periods. Rather, the study of the titles of the high priest points to a much more complex and intricated history, involving a mix of continuous and discontinuous elements between these periods. Further studies of the history of the high priest in the pre-Hasmonean period will need to take that complexity into account.

**Funding:** This research received no external funding.

**Conflicts of Interest:** The author declares no conflict of interest.

## Notes

[1] Apart from a golden bell found on the site of the Gerizim temple and which may have belonged to the vestment of the high priest (see Magen et al. 2004, and compare with Exod 28:33–35), we do not have direct evidence from Samaritan sources for the high priest of Samaria in the pre-Hasmonean period. External accounts, such as Josephus' narrative of the establishment of the Samaritan cult on Mount Gerizim (*Ant.* 11.306–312), are clearly polemical and hardly comprise a reliable source of information. The extent to which it is possible to reconstruct the history of the Samarian/Samaritan high priesthood until the destruction of the sanctuary on Mount Gerizim by Johannes Hyrcanus is a topic that deserves a distinct study.

[2] For instance, the book of Nehemiah consistently refers to the high priest Eliašib using the title הכהן הגדול (Neh 3:1, 20; 13:28). In one passage (Neh 13:4), a priest named Eliašib is simply called "the priest" but this seems to be precisely because this Eliašib is not the high priest of Jerusalem, but another priest tasked with the supervision of the rooms inside the temple. See, e.g., (Blenkinsopp 1988, pp. 353–54).

[3] According to (Gonzalez and Mendoza 2020), the termination of the office of *pēḥ*â, or "governor", goes back to Ptolemy's campaign in the southern Levant from 311 BCE.

[4] Wöhrle would actually ascribe the entirety of the account in 2 Kgs 25 to an early postexilic edition of 1–2 Kings, which he dates to the end of the 6th century BCE (Wöhrle 2008, pp. 228–30). However, this issue need not be further discussed here.

[5] I am grateful to the anonymous reader who reminded me of this point.

6     Note that according to Frevel 2013, p. 160, Lev 21:10 is a late addition to H based on Lev 10:6–7, which would place the whole introduction of the title הכהן הגדול in the Hexateuch still later; however, the relationship between Lev 10:6–7 and Lev 21:10 is more likely the reversed one, see (Nihan 2019, pp. 228–30).

7     V. 4b, which clearly interrupts the transition between v. 4a and 5, should be a still later interpolation; compare, e.g., Levin 2003, pp. 207, 208 n. 46.

8     Jer 51:59 MT (28:59 LXX) mentions a "Seraiah," son of Neriah and grandson of "Maḥseiah," but this Seraiah is arguably distinct from the high priest of Jerusalem since he is designated as an "officer of rest"—an expression presumably referring to a quartermaster of sorts.

9     See *FGH* III.A 264, F6. Book 40 of Diodorus Siculus' History is lost to us, and is known only through the paraphrase provided by Photius (9th century CE). For the edition of Photius, see Henry 1971.

10    The only exception would be Ezra 7:5, which ascribes the title כהן הראש to Aaron. However, this may represent a later extension of the usage documented in 1–2 Chronicles.

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
