# Peer review of "The Titles of the High Priest of Jerusalem in the Pre-Hasmonean Period"

_religions, doi:10.3390/rel14040529_

Round 1

Reviewer 1 Report

The article is well-written, well-argued and clarifies an interesting, relevant problem. Thus, one should look forward to its publication.

There is only one aspect that needs rephrasing–the role of Lev 21:10 for the argument. As commented upon in lines 550–52, the phrase there is not a title, strictly (syntactically) speaking. One can surely argue that the passage plays with the title and combines different notions etc., but it should be made clear from the outset that this passage differs significantly from the other attestations. As a minimally invasive solution I suggest a footnote at the earliest mention of the passage referring to lines 550–52. However, a slightly more detailed treatment would be preferable.

Minor corrections should include:

l. 39 “construct” – that is misunderstandable as grammatical technical term, “phrase” or “construction” seem preferable

ll. 45/47 2Chr 31:10 is counted twice, thus the actual number is smaller (see below)

ll. 47/48 the expression הכהן ראש is actually also ungrammatical according to standard BH grammar

ll. 82/83 and l. 126 should be “five out of eight” (see above)

ll. 190 “form” → “from”

ll. 237ff. it would be apt to remark that vv. 4–6 are missing in Cod. Vaticanus and thus, presumably, in the OG (cf., e.g., the discussion Reinhard Müller, Juha Pakkala, Bas ter Haar Romeny: Evidence of Editing: Growth and Change of Texts in the Hebrew Bible. SBL Resources for Biblical Study 75. Atlanta 2014, 45–58).

l. 331 ברוך בכור (there are a few other places where the blanks around the Hebrew are misplaced!)

l. 398 “ore” → “more”

l. 403 “Escurus” → “Excursus”

l. 404 “detailed” → “details”

l. 420 maybe cut “at best” to make it less sharp?

Author Response

Many thanks for your comments. I have included all your corrections and suggestions. In particular, I have rewritten the section on Lev 21:10 (now lines 568-580) and included a brief comment when the paper first mentions this vers (lines 39-41). I have also added two sentences regarding Josh 20:4-6 in the OG (lines 257-260).

Thanks again. These were all helpful comments.

Reviewer 2 Report

The article's argumentation is clear and convincing. The author shows a fine knowledge of relevant scholarship.

I would appreciate a short discussion on the historical implications of the development of these high priestly terms. What can be added that we did not know before and the history of this office? 

Notes:

Dating the Holiness Code. Knohl and Milgrom dated it earlier, to the preexilic period.

Typos: l.331: should be משנה/בכור   not ברוך

l. 642: Use of the Word         as – something is missing here?

Author Response

Many thanks for your comments. I have included your corrections, and added a reference to Knohl's and Milgrom's dating of H (now lines 227-230). Finally, I have added a paragraph in the conclusion (lines 644-661) whre I discuss the implications of this study from a more historical perspective.